# Molecular Typing, Antibiotic Resistance and Enterotoxin Gene Profiles of *Staphylococcus aureus* Isolated from Humans in South Korea

**DOI:** 10.3390/microorganisms10030642

**Published:** 2022-03-17

**Authors:** Sunghyun Yoon, Yon Kyoung Park, Tae Sung Jung, Seong Bin Park

**Affiliations:** 1National Center for Toxicological Research, U.S. Food and Drug Administration, Jefferson, AR 72079, USA; sunghyunyoon0707@gmail.com; 2Microbiology Division, Busan Institute of Health and Environment, Busan 616-100, Korea; akacia@korea.kr; 3Laboratory of Aquatic Animal Diseases, Research Institute of Life Science, College of Veterinary Medicine, Gyeongsang National University, Jinju 660-701, Korea; jungts@gnu.ac.kr; 4Coastal Research & Extension Center, Mississippi State University, Pascagoula, MS 39567, USA

**Keywords:** *S. aureus*, antimicrobial susceptibility, enterotoxin gene, MALDI-TOF MS, RAPD

## Abstract

The emergence of antimicrobial-resistant *Staphylococcus aureus* has become a grave concern worldwide. In this study, 95 strains of *S. aureus* isolated from stool samples were collected from Busan, South Korea to characterize their antimicrobial susceptibility, enterotoxin genes, and molecular typing using matrix-assisted laser desorption/ionization time-of-flight mass spectrometry (MALDI-TOF MS) and random amplification of polymorphic DNA (RAPD) assay. Only two strains showed no drug resistance, whereas resistance to three or more antibiotics was observed in 87.4% of strains. Ampicillin resistance was the most common at 90% and all strains were susceptible to vancomycin. The distribution of enterotoxin genes encoded in isolates was sea (32.6%), sec (11.6%), seg (19%), sea & sec (2.1%), and sec & seg (34.7%). Molecular typing using both MALDI-TOF MS and RAPD indicated that *S. aureus* exhibited diverse clonal lineages and no correlations were observed among the profiling of enterotoxin, MALDI-TOF MS, and RAPD. This investigation provides useful information on foodborne pathogenic *S. aureus* that has a significant public health impact in South Korea.

## 1. Introduction

*Staphylococcus aureus* is a round, Gram-positive, facultative anaerobic bacterium that is part of the natural microflora of humans. However, this bacterium is also a pathogen associated with a number of diseases; in particular, it is the causative agent of staphylococcal enteritis (i.e., staphylococcal food poisoning), which is characterized by gastroenteritis, vomiting, diarrhea, abdominal pain, etc. [1]. Among many diseases by toxins including food poisoning and toxic shock syndrome, this toxin-mediated food poisoning is brought about by the enterotoxins produced by *S. aureus*. While proliferating in foods such as dairy, meat, eggs and vegetables, *S. aureus* releases more than 20 different staphylococcal enterotoxins (SE) toxins. Of them, SEA and SED are the most common toxins in staphylococcal food poisoning worldwide [2,3]. These toxins are thermostable and resistant to stomach proteases, and ingestion of *S. aureus*-contaminated food can be fatal. Moreover, spoiled or contaminated food may not show signs of spoilage (e.g., changes in odor, color and/or flavor), which generates public health threats [4].

The emergence of multidrug-resistant *S. aureus* is also a major concern in public health since *S. aureus* can acquire plasmids or transposons encoded with antimicrobial-resistant genes from other species and genera [5]. The spread of antimicrobial-resistant *S. aureus* such as methicillin-resistant *S. aureus* (MRSA) is one of the major problems in health care settings [6].

When a patient is suspected of suffering from foodborne illness, the conventional diagnosis usually consists of culturing a stool sample on selective agar or nutrient agar, followed by identification of the causative bacterium using biochemical tests, such as Gram staining and catalase, oxidase, and API tests [7,8]. These procedures are simple and inexpensive, but they can be laborious and time-consuming. Thus, a number of molecular biology techniques have been adapted for the rapid identification of some bacterial pathogens. For the detection of the SE toxins of *S. aureus*, for example, polymerase chain reaction (PCR) assays, enzyme-linked immunosorbent assays (ELISAs) and agglutination tests are recommended [2,4]. More recently, matrix-assisted laser desorption ionization time-of-flight mass spectrometry (MALDI-TOF MS), wherein the spectra of unknown microorganisms are compared with those of a reference microorganism, has shown promise for the rapid, accurate and inexpensive identification of bacteria [9].

Diverse methods have been used to assess the molecular typing and analysis of *S. aureus* isolates such as random amplified polymorphism DNA (RAPD) PCR [10], and MALDI-TOF MS [11]. These tools are considered useful for comparison of various types of bacteria in research laboratories [11].

Many studies have reported that *S. aureus* could transfer their antimicrobial resistances as well as virulent genes to adjacent bacteria through their mobile genetic elements [12]. Therefore, *S. aureus* in contaminated foods and feces could contribute to the spread of antimicrobial resistances and virulence to humans [1]. The aim of the present study is to demonstrate the distribution of antimicrobial resistance and virulence factors in *S. aureus* strains recovered from human stool samples in Korea. Further correlation of *S. aureus* strains was analyzed by two molecular typing methods, RAPD and MALDI-TOF.

## 2. Materials and Methods

### 2.1. S. aureus Isolation and Growth Conditions

Stool specimens were collected from 152 patients suffering from foodborne diarrheal diseases from 2014 to 2016 in Busan, South Korea. Each stool specimen was grown on Baird–Parker (BP) agar (Oxoid, Hampshire, UK) with 5% egg-yolk tellurite emulsion (Oxoid) at 35 °C overnight. Small black colonies with transparent zones on BP agar were selected and sub-cultured on blood agar plates (BAP) at 35 °C overnight. Gram staining, the catalase test, and the coagulase test were carried out, and biochemical tests were performed with an API 20 Staph kit (BioMerieux, Durham, NC, USA) to confirm that the isolates were *S. aureus*.

### 2.2. PCR-Based Detection of Enterotoxin Genes

Genomic DNA was extracted from the *S. aureus* isolates by the boiling method [13]. Briefly, 1 mL of *S. aureus* cultured overnight in brain heart infusion broth (BHI, Oxoid) was centrifuged at 9000× *g* for 3 min. The pellet was suspended with 1 mL of PBS and centrifuged at 9000× *g* for 3 min. The washed pellet was suspended in 100 μL of sterile distilled water, boiled for 20 min, and centrifuged at 14,000× *g* for 10 min. The supernatant was used as the template for PCR. Two multiplex PCR methods designated sets A and B were used to detect the sea, seb, sec, sed, see and seg genes (Appendix A). The Set A reaction contained 10 pmol each of the sea, seb and sec primer pairs, whereas the set B reaction contained 10 pmol each of the sed, see and seg primer pairs. The sequences of the primers are given in Appendix A. Each multiplex PCR mixture included 2.5 units of i-StarMAXTM DNA polymerase (INTRON, Daejeon, Korea), 10 mM dNTP (2.5 mM each), 1 × PCR buffer, 2 μL of each primer set (10 pmol), and 2 μL of bacterial DNA. Distilled water was added to a final volume of 20 μL. PCR was performed using a T-100™ programmable thermal controller (Bio-Rad, Irvine, CA, USA) and the following conditions: one cycle of 94 °C for 5 min, 35 cycles of 95 °C for 1 min, 53 °C for 1 min and 72 °C for 1 min, and a final extension at 72 °C for 5 min. The amplified PCR products were visualized on 1.5% TAE agarose gels containing 0.5 μg/mL ethidium bromide.

### 2.3. Antimicrobial-Agent Susceptibility Test

The antibiotic resistance of *S. aureus* isolates was determined using the standard disk diffusion method described by the Clinical and Laboratory Standards Institute (CLSI) [14], with application of ampicillin (10 μg), chloramphenicol (30 μg), ciprofloxacin (5 μg), cefepime (30 μg), cefotetan (30 μg), clindamycin (2 μg), erythromycin (15 μg), gentamicin (10 μg), imipenem (10 μg), oxacillin (1 μg), penicillin (10 U), rifampin (5 μg), trimethoprim/sulfamethoxazole (1.25 μg/23.75 μg), tetracycline (30 μg) and vancomycin (30 μg). The isolates were spread on tryptone soya agar (TSA, Oxoid) and cultured at 35 °C for 24 h. Colonies were suspended in 3 mL of Mueller–Hinton broth (Oxoid), and the suspension turbidity was adjusted to 0.5 McFarland standard (BioMerieux, Durham, NC, USA). The bacterial solution was then evenly spread on Mueller–Hinton agar (Oxoid, UK), and antimicrobial susceptibility test disks (BD, Franklin Lakes, NJ, USA) were placed on the plates. After the plates were incubated for 24 h at 35 °C, clear zones were examined, and an electronic digital caliper (Fisher Scientific, Hampton, NH, USA) was used to measure the growth inhibition of the isolates in response to each antimicrobial agent. The susceptibilities of the isolates were determined based on the standard suggested by CLSI [14]. *E. coli* ATCC 25922 and *S. aureus* ATCC 29213 were used as control strains.

### 2.4. MALDI-TOF Mass Spectrometry

#### 2.4.1. Sample Preparation

The direct colony and standard extraction methods were used to prepare the isolates for MALDI biotyping analysis as the manufacturer recommended. In brief, in the direct colony method, fresh bacterial colonies were applied directly onto an MSP 96 target polished steel plate (Bruker Daltonik, Bremen, Germany), air dried, mixed with 1 μL of HCCA matrix solution (a saturated solution of α-cyano-4-hydroxy-cinnamic acid in 50% acetonitrile and 2.5% trifluoroacetic acid) to crystallize the sample, and air dried.

#### 2.4.2. MALDI-TOF MS

Mass spectra were obtained using a Microflex LT mass spectrometer (Bruker Daltonik) controlled by the Flexcontrol software (Version 3.0; Bruker Daltonik). Positive ions were extruded with an accelerating voltage of 20 kV and the spectra were analyzed within a mass/charge (*m*/*z*) ratio of 2000 to 20,000 in the positive linear mode. Each spectrum was calibrated with a bacterial test standard (BTS 255343, Bruker Daltonik). The generated spectra were automatically matched with the reference library and scored using integrated pattern-matching algorithm software (MALDI Biotyper RTC, Bruker Daltonik). Logarithmic scores of 0 to 3 were assigned according to the matching patterns of the spectral peaks. Scores of 0 to 1.699 indicated no reliable identification; 1.700 to 1.999 indicated a probable genus-level identification; 2.000 to 2.299 indicated a secure genus-level identification, probable species-level identification; and scores of 2.300 to 3.000 indicated a highly probable species-level identification. A main spectra library (MSP) dendrogram was generated using the MALDI Biotyper 3.0, with the distance level in the dendrogram set to a maximal value of 1000, as recommended by the manufacturer.

### 2.5. Random Amplified Polymorphic DNA (RAPD) Analysis

Chromosomal DNA was extracted with an Accuprep genomic DNA extraction kit (Bioneer, Daejeon, Korea) following the manufacturer’s protocol. Random amplified polymorphic DNA (RAPD) analysis was performed using Ready-To-Go RAPD Analysis Beads (GE Healthcare, Piscataway, NJ) according to the manufacturer’s instructions. Briefly, each 25 μL reaction volume contained 10 ng of template DNA and 25 pmol of RAPD analysis primer 5 (5′-d(AACGCGCAAC)-3′), which was found to generate unique banding patterns in our preliminary experiments (data not shown). Amplification was performed using a PT-100TM thermocycler (MJ Research, Watertown, MA, USA) and the following protocol: one cycle at 95 °C for 5 min, 45 cycles of 95 °C for 1 min, 36 °C for 1 min, and a final incubation at 72 °C for 2 min. The resulting PCR products were visualized as described above, and the results were analyzed using the Bionumerics software (Applied Maths, Austin, TX, USA) to generate the phylogenetic tree.

## 3. Results

### 3.1. Detection of Enterotoxins

A total of 95 *S. aureus* isolates were obtained from stool samples of diarrheal patients, and their identifications as *S. aureus* were confirmed through biochemical tests. Only one isolate per person was included for further analysis. Multiplex PCR assays of enterotoxin genes revealed that all of these foodborne *S. aureus* isolates harbored one or more of the genes—sea, sec and seg—but none of the isolates harbored the seb, sed or see genes (Table 1).

### 3.2. Antimicrobial-Agent Susceptibility Test

The *S. aureus* isolates were highly resistant to ampicillin (94.7%) and penicillin (95.8%); moderately resistant to cefepime, cefotetan, chloramphenicol, ciprofloxacin, clindamycin, erythromycin, gentamicin, imipenem, oxacillin, tetracycline, and trimethoprim/sulfamethoxazole; and minimally resistant to rifampin (4.2%). All of the isolates were susceptible to vancomycin (Table 2). Ninety-two strains (96.8%) were multidrug-resistant. Two of the strains were resistant to 13 of the tested antibiotics. Among the multidrug-resistant isolates, many were resistant to 11 antibiotics (44.2%). Only one strain was resistant to just a single antibiotic (Table 3).

### 3.3. RAPD Analysis

RAPD analysis was performed using RAPD primer 5 of the Ready-To-Go-RAPD Analysis kit. This primer was chosen from the six provided primers because 20 preliminary tests showed that it could generate clear and diverse amplification patterns (data not shown). The same amplification patterns were observed in three independent experiments. Unweighted pair group method with arithmetic mean (UPGMA) clustering was performed using the Bionumerics software, and identified six distinct groups/clusters of *S. aureus* isolates at a genetic distance of 78 (Figure 1).

### 3.4. MALDI-TOF MS

All of the *S. aureus* isolates were identified to the species level (log scores ≥ 2.0) using both the direct colony and standard extraction methods. The generated MSP dendrogram yielded five groups/clusters (Figure 2). We failed to observe any correlation in the clusters of our RAPD analysis and MSP dendrogram.

## 4. Discussion

Staphylococci are ubiquitous, can be isolated from food products, and are responsible for a number of animal and human diseases [15]. The enterotoxins released by *S. aureus* are responsible for the bacterial food poisoning caused by this pathogen, with the SEA and SED pathotypes (i.e., strains expressing the sea and/or sed genes) highly associated with disease [16,17]. To detect the pathogen and identify its expressed toxin(s), a multiplex PCR assay is usually recommended [4,16,17]. A previous study on 430 *S. aureus* isolates obtained from dairy products found eight SEA strains, three SEB strains, and two SED strains [4]. In several studies on isolates obtained from food, SEA, SEB, and SED were found to be the major enterotoxins [4,18,19]. In the present study, in contrast, multiplex PCR revealed that the most dominant toxins expressed in the 95 isolated strains were SEC and SEG (33 strains), followed by SEA (31 strains), SEG (18 strains), SEC (11 strains) and SEA plus SEC (2 strains). This apparent discrepancy may reflect differences in number and origin of the samples/isolates.

MALDI-TOF MS using both the direct colony and standard extraction methods confirmed the identification of our *S. aureus* isolates (100%). An earlier study demonstrated that 94% of *S. aureus* isolates can be identified to the genus level using the standard extraction method [9]. Another report showed that MALDI-TOF MS could perform species-level detection for eight of 20 *S. aureus* isolates [20]. The 100% success rate obtained in the present study suggests the importance of using freshly cultured isolates, and the advantage of using the latest software for bacterial isolation. Additionally, the use of MALDI-TOF MS is accurate, inexpensive, faster and requires less expertise compared to conventional microbiological methods such as spa-typing or MLST-typing or time consuming, expensive methods such as whole genome sequencing [11].

Earlier studies showed that the results from MALDI-TOF MS and other clustering analyses (e.g., RAPD) may show correlations based on their hosts and geographic origins [21,22,23]. However, we failed to find any correlation between the MALDI-TOF MS dendrogram and RAPD fingerprints of our *S. aureus* isolates. Similarly, a previous study failed to find any correlation between the typing patterns obtained from RAPD and MALDI-TOF MS of various Enterococci [24]. In our case, the lack of correlation may reflect that all of the isolates originated from a single city (Busan, South Korea) and host (human).

Antibiotics are often used to treat foodborne diseases [25], but their indiscriminate and unmonitored use has led to drug resistance in certain bacterial species. Here, we show that all of the obtained foodborne *S. aureus* isolates were resistant to at least one antibiotic, and that 96.8% were resistant to multiple antibiotics. Most of the isolates showed resistance to ampicillin (94.7%) and penicillin (95.8%). A recent study also showed that 70% of the tested *S. aureus* strains were resistant to ampicillin and penicillin [26]. The resistance to oxacillin (77.9%) was noteworthy, which contains public health concerns such as the emergence of methicillin-resistant *S. aureus* (MRSA) [27]. Fortunately, all of the isolates were found to be susceptible to vancomycin.

## 5. Conclusions

Here, we analyzed the characteristics of *S. aureus* isolated from the stool samples of diarrheal patients. We found that all of the isolated *S. aureus* strains possessed at least one enterotoxin gene, and most were multidrug-resistant. We were also able to show that MALDI-TOF MS can be a practical and faster method for identifying bacterial isolates using either the direct colony or standard extraction methods. These results may assist in the establishment of treatment protocols for *S. aureus*-mediated illnesses.

## Figures and Tables

**Figure 1 microorganisms-10-00642-f001:**
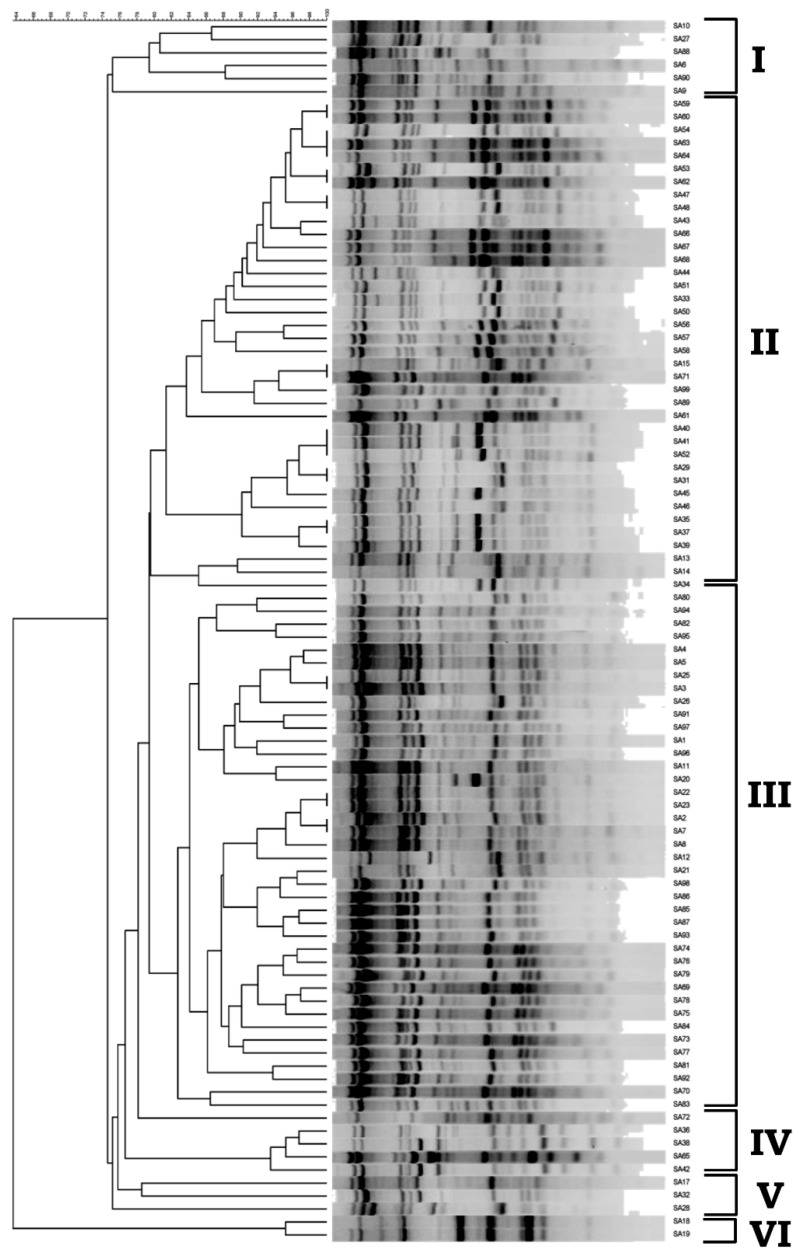
UPGMA clustering of the random amplified polymorphic DNA (RAPD) profiles of *S. aureus* isolates. Profiles were generated by RAPD primer 5 (Ready-To-Go-RAPD Analysis kit) and the UPGMA clustering (I to VI) was generated using Bionumerics.

**Figure 2 microorganisms-10-00642-f002:**
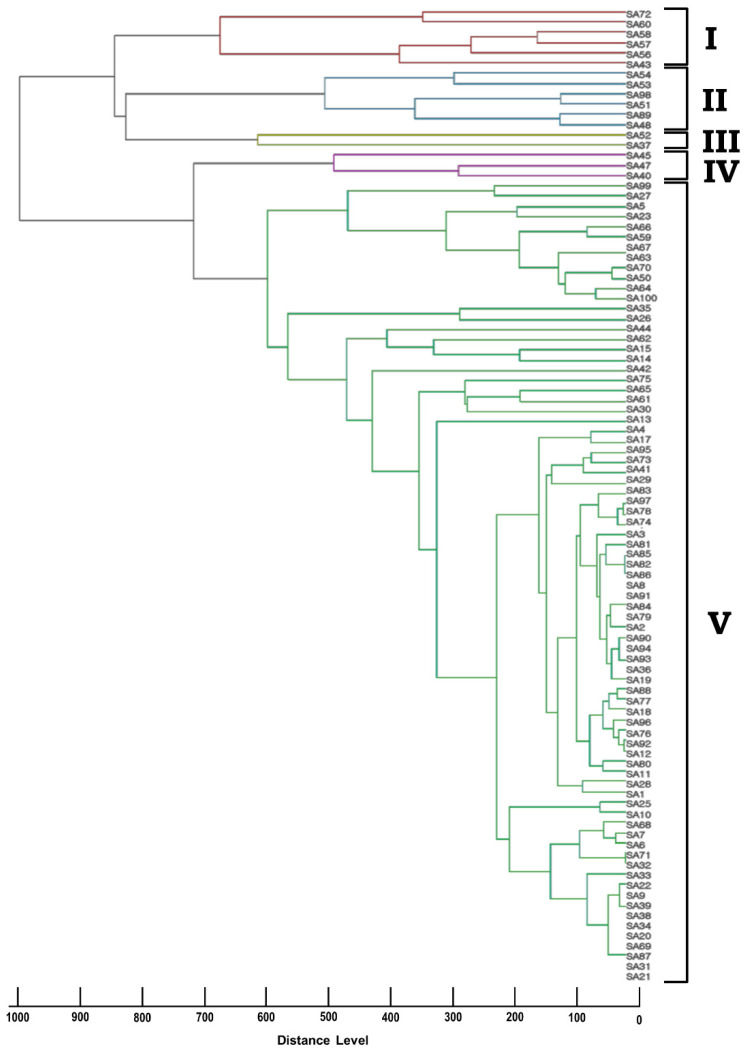
Cluster analysis of the MSP dendrograms generated by the MALDI Biotyper 3.0 for *S. aureus* isolates. The MSP dendrogram results are clustered as groups (I to V) based on their distance levels. The distance level in the dendrogram was set at a maximal value of 1000.

**Table 1 microorganisms-10-00642-t001:** Distribution of enterotoxin genes in the 95 *S. aureus* strains.

Enterotoxin Gene(s)	No. of Isolates Positive for Gene(s) (%)
*sea*	31 (32.6)
*sec*	11 (11.6)
*seg*	18 (18.9)
*sea* & *sec*	2 (2.1)
*sec* & *seg*	33 (34.7)

**Table 2 microorganisms-10-00642-t002:** Antimicrobial-agent resistance profiles of the 95 *S. aureus* strains.

Antimicrobial Agent	No. of Resistant Isolates (%)	No. of Intermediate Isolates (%)	No. of Susceptible Isolates (%)
Ampicillin	90 (94.7)	0 (0)	5 (5.3)
Cefepime	73 (76.8)	1 (1.1)	21 (22.1)
Cefotetan	66 (69.5)	3 (3.2)	26 (27.3)
Chloramphenicol	29 (30.5)	1 (1.1)	65 (68.4)
Ciprofloxacin	41 (43.2)	0 (0)	54 (56.8)
Clindamycin	56 (58.9)	2 (2.1)	37 (39.0)
Erythromycin	71 (74.7)	4 (4.2)	20 (21.1)
Gentamicin	72 (75.8)	1 (1.1)	22 (23.1)
Imipenem	68 (71.6)	0 (0)	27 (28.4)
Oxacillin	74 (77.9)	0 (0)	21 (22.1)
Penicillin	91 (95.8)	0 (0)	4 (4.2)
Rifampin	4 (4.2)	2 (2.1)	89 (93.7)
Tetracycline	68 (71.6)	1 (1.1)	26 (27.3)
Trimethoprim/sulfamethoxazole	39 (41.1)	3 (3.2)	53 (55.0)
Vancomycin	0 (0)	0 (0)	95 (100.0)

**Table 3 microorganisms-10-00642-t003:** Antimicrobial-agent resistance patterns of the 95 *S. aureus* stains.

No. of Antimicrobials	Resistance Pattern	No. of Isolates
0	-	2
1	GM	1
2	GM, E	1
	AM, P	6
	OX, P	1
	AM, P	1
3	AM, FEP, P	1
	AM, P, E	3
	AM, SXT, P	2
	AM, GM, P	1
	AM, OX, P	1
4	AM, P, RA, E	1
	AM, FEP, OX, P	1
	AM, GM, TE, P	1
5	AM, OX, P, E, CC	1
	AM, FEP, SXT, OX, P	1
7	AM, FEP, TE, C, P, E, CC	1
9	AM, GM, FEP, CTT, CIP, IPM, OX, P, E	1
	AM, GM, FEP, CIP, IPM, TE, OX, P, E	1
10	AM, GM, FEP, CTT, CIP, IPM, OX, P, E, CC	1
	AM, FEP, CTT, IPM, SXT, C, TE, OX, P, CC	1
	AM, GM, FEP, CTT, IPM, SXT, C, TE, OX, P	2
	AM, GM, FEP, CTT, CIP, IPM, TE, OX, P, CC	1
	AM, GM, FEP, CTT, CIP, IPM, TE, P, E, CC	1
	AM, GM, FEP, IPM, SXT, TE, OX, P, E, CC	2
11	AM, GM, FEP, CTT, IPM, SXT, C, TE, OX, P, E	1
	AM, GM, FEP, CTT, CIP, IPM, SXT, OX, P, E, CC	1
	AM, GM, FEP, CTT, IPM, SXT, TE, OX, P, E, CC	1
	AM, GM, FEP, CTT, SXT, C, TE, OX, P, E, CC	1
	AM, GM, FEP, CTT, IPM, SXT, C, TE, OX, P, E	11
	AM, GM, FEP, CTT, CIP, IPM, TE, OX, P, E, CC	25
12	AM, GM, FEP, CTT, CIP, IPM, TE, OX, P, RA, E, CC	2
	AM, GM, FEP, CTT, CIP, IPM, C, TE, OX, P, E, CC	1
	AM, GM, FEP, CTT, CIP, IPM, SXT, TE, OX, P, E, CC	5
	AM, GM, FEP, CTT, IPM, SXT, C, TE, OX, P, E, CC	9
13	AM, GM, FEP, CTT, CIP, IPM, SXT, TE, OX, P, RA, E, CC	1
	AM, GM, FEP, CTT, CIP, IPM, SXT, C, TE, OX, P, E, CC	1
	37 patterns	95

Abbreviations: GM, gentamicin; E, Erythromycin; AM, ampicillin; P, Penicillin; OX, Oxacillin; FEP, Cefepime; SXT, Trimethoprim/Sulfamethoxazole; RA, Rifampin; TE, Tetracycline, CC, Clindamycin; C, Chloramphenicol; CTT, Cefotetan; CIP, Ciprofloxacin; IPM, Imipenem.

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
