# Peer review of "Molecular Typing, Antibiotic Resistance and Enterotoxin Gene Profiles of Staphylococcus aureus Isolated from Humans in South Korea"

_microorganisms, 2022, doi:10.3390/microorganisms10030642_

Round 1
Reviewer 1 Report
Yoon et al. report on the prevalence of enterotoxin genes in S. aureus isolates from stool samples of Korean patients. So, they contribute to available epidemiological knowledge. However, I have a number of suggestions.
1.) The applied typing approaches are outdated, more precise and better comparable/standardizable techniques are available. Regarding molecular typing, at least spa-typing or MLST-typing, better whole genome sequencing, would have provided better international comparability. Instead of simple MALDI-TOF-MS, a MALDI-based proteotyping approach might have yielded more detailed information. I agree that it may be unfeasible to add such additional assessments with acceptable effort for the authors, but they should at least discuss such methodical limitations in the discussion of their manuscript.
2.) Introduction: When the authors introduce S. aureus-associated enterotoxins, it should at least be mentioned that other virulence factors exist as well; suitable reviews should be quoted. The sudden switches in the introduction from toxins via resistance to diagnostic tests seems arbitrary. The introduction should show a logical composition leading to the aim of the study. If they introduce diagnostic approaches at all, the authors should also include up-to-data procedures.
3.) Introduction, lines 66-67: Although fecal-oral transmission of S. aureus in human patients is possible, it is hardly the most important mode of transmission.
4.) Methods, line 81: If the authors introduce procedures like “the boiling method”, at least a reference should be provided.
5.) Methods, line 90: The primers are provided in the Table S1, not in the Table 1 as stated in line 90.
6.) Methods, sample preparation for MALDI-TOF-MS) Are there any deviations from the manufacturer-recommended protocol which justify such a detailed description? If the authors just applied the manufacturer’s standard procedure, it may be sufficient to appropriately reference it.
7.) Results, table 2: Is there a specific reason why interpretative reading of resistance profiles was not performed? If yes, please explain.
8.) Discussion, line 220: The authors have quoted a pretty old study here. In the mean time, MALDI-TOF-MS has become a standard diagnostic procedure in many diagnostic microbiological laboratories with a very high reliability regarding the identification of common species like S. aureus.
9.) Lines 235-238: The authors interpretation reads as if they were surprised by the detected proportion of penicillinase carriage as indicated by penicillin and ampicillin resistance. However, the reported oxacillin-resistance rate of more than 75% is the much more striking finding requiring broader discussion. Have the author confirmed this, e.g., by screenings for mecA or mecC genes? Is this finding in line with previous regional findings?
Author Response
Point-to point responses to review no.1
Yoon et al. report on the prevalence of enterotoxin genes in S. aureus isolates from stool samples of Korean patients. So, they contribute to available epidemiological knowledge. However, I have a number of suggestions.
1.) The applied typing approaches are outdated, more precise and better comparable/standardizable techniques are available. Regarding molecular typing, at least spa-typing or MLST-typing, better whole genome sequencing, would have provided better international comparability. Instead of simple MALDI-TOF-MS, a MALDI-based proteotyping approach might have yielded more detailed information. I agree that it may be unfeasible to add such additional assessments with acceptable effort for the authors, but they should at least discuss such methodical limitations in the discussion of their manuscript.
-> Thank you so much for the precious suggestions. I agree that there are other methods could be used. We will do spa-typing and MLST-typing in our next study, and also, we are planning to do whole genome sequencing for the S. aureus strains next time. The explanation for other methods and limitation in this study is added in discussion part. (L221: … methods such as spa-typing or MLST-typing or time consuming, expensive method such as whole genome sequencing.)
2.) Introduction: When the authors introduce S. aureus-associated enterotoxins, it should at least be mentioned that other virulence factors exist as well; suitable reviews should be quoted. The sudden switches in the introduction from toxins via resistance to diagnostic tests seems arbitrary. The introduction should show a logical composition leading to the aim of the study. If they introduce diagnostic approaches at all, the authors should also include up-to-data procedures.
-> The introduction part was changed and added some explanation for better reading in terms of toxins and antimicrobial resistance as suggested.
(L34-35) Among many diseases by toxins including food poisoning and toxic shock syndrome, …
(L40-42) … contaminated food may not show signs of spoilage which gives public health threats [4]. The emergence of multidrug resistant S. aureus is also the major concern in public health
(L221) such as spa-typing or MLST-typing or time consuming, expensive method such as whole genome sequencing.
3.) Introduction, lines 66-67: Although fecal-oral transmission of S. aureus in human patients is possible, it is hardly the most important mode of transmission.
-> The sentence is changed as suggested (Therefore, S. aureus in contaminated foods and feces could contribute… [1]).
4.) Methods, line 81: If the authors introduce procedures like “the boiling method”, at least a reference should be provided.
-> Reference is added as suggested [13]. (Lee et al., 2009)
5.) Methods, line 90: The primers are provided in the Table S1, not in the Table 1 as stated in line 90.
-> Changed as suggested (L88: Table 1 -> Table S1).
6.) Methods, sample preparation for MALDI-TOF-MS) Are there any deviations from the manufacturer-recommended protocol which justify such a detailed description? If the authors just applied the manufacturer’s standard procedure, it may be sufficient to appropriately reference it.
-> Changed the part as suggested for brief explanation. (Deleted detailed explanation L122-129)
7.) Results, table 2: Is there a specific reason why interpretative reading of resistance profiles was not performed? If yes, please explain.
-> The explanation of antimicrobial resistance is from L230-238. The more detailed tests (antimicrobial gene detection such as mecA or mecC) will be performed in the next study and the multi drug resistance (MDR) and resistance of importance such as MRSA will be discussed.
8.) Discussion, line 220: The authors have quoted a pretty old study here. In the mean time, MALDI-TOF-MS has become a standard diagnostic procedure in many diagnostic microbiological laboratories with a very high reliability regarding the identification of common species like S. aureus.
-> Changed the paragraph to show not only the MALDI-TOF MS is accurate but also affordable and faster method compared to others (L219-L222: Also, the use of MALDI-TOF MS is accurate, inexpensive, faster and requires less expertise compared to conventional microbiological methods such as spa-typing or MLST-typing or time consuming, expensive method such as whole genome sequencing [11].).
9.) Lines 235-238: The authors interpretation reads as if they were surprised by the detected proportion of penicillinase carriage as indicated by penicillin and ampicillin resistance. However, the reported oxacillin-resistance rate of more than 75% is the much more striking finding requiring broader discussion. Have the author confirmed this, e.g., by screenings for mecA or mecC genes? Is this finding in line with previous regional findings?
-> added part for the explanation of oxacillin-resistance (L235-237: The resistance to oxacillin (77.9%) was noteworthy which contains public health concerns such as the emergence of methicillin-resistant S. aureus (MRSA) [26].). The mecA or mecC were not tested in this study but we are planning of testing them and positive MRSA strains will be tested further for the gene mutation or next generation sequencing (NGS).
Reviewer 2 Report
The work was well organised and executed, and the manuscript is to the point and concise. I have three main questions and a couple of editorial comments, answering of which should result in the publication of the manuscript in Microorganism.
Introduction
Line 54, … assays (ELISAs)
Materials and methods
Line 82, what was the (approximate) starting amount of Staphylococcus aureus cell mass of genomic DNA extraction? Was the CFU/mL determined to that a link to the 50 mL can be made (50 mL as such do not mean a lot).
Lien 90, Table S1
Line 107, “adjusted to 0.5 McFarlane” shall be defined in more detail (seems to be jargon)
Line 118, biotyping?
Line 152 and 183, data now shown
Results
Line 164, how can it be explained that none of the isolates from stool samples of humans suffering food poisoning contains the sed gene, whereas SEA and SED are the most common toxins in staphylococcal food poisoning outbreaks (line 37 and references 2 and 3)?
Line 169, moderately resistant, why is only erythromycin mentioned here? I think all antibiotics other than ampicillin, penicillin and rifampin have to be mentioned here (moderate resistance, 31 – 78 %).
Line 175, please delete current text
Discussion
Line 215, how can the absence of a correlation be explained with the composition of the multiplex PCR method?
Line 222, MALDI-TOF MS
Conclusions
Line 240, We analysed
Line 245, illnesses.
References
Please indicated the year in bold, and please indicate the abbreviated journal name in italic
Supplementary material
Table S1, text below table: References; please indicate only the journal volume, not the issue.
Author Response
Point-to point responses to review no.2
The work was well organised and executed, and the manuscript is to the point and concise. I have three main questions and a couple of editorial comments, answering of which should result in the publication of the manuscript in Microorganism.
Introduction
Line 54, … assays (ELISAs)
-> Changed as suggested (added s)
Materials and methods
Line 82, what was the (approximate) starting amount of Staphylococcus aureus cell mass of genomic DNA extraction? Was the CFU/mL determined to that a link to the 50 mL can be made (50 mL as such do not mean a lot).
-> The sentences are changed for better explanation.
(50 μl of S. aureus cultured overnight in brain heart infusion broth (BHI, Oxoid) were mixed with 450 μl of sterile distilled water, washed twice with PBS, and centrifuged at 9,000 × g for 3 min.
-> 1mL of S. aureus cultured overnight in brain heart infusion broth (BHI, Oxoid) were centrifuged at 9,000 × g for 3 min. The pellet was suspended with 1mL of PBS and centrifuged for at 9,000 × g.)
Lien 90, Table S1
-> Changed as suggested (Table 1 -> Table S1)
Line 107, “adjusted to 0.5 McFarlane” shall be defined in more detail (seems to be jargon)
-> Changed the phrase for better understanding (…suspension was adjust to 0.5 McFarland -> suspension turbidity was adjusted to 0.5 McFarland standard). McFarland standards are used as a reference to adjust the turbidity of bacterial suspensions and 0.5 McFarland indicates approximately 1.5 X 10^8 CFU/mL.
Line 118, biotyping?
-> Changed the word as suggested (biotyper -> biotyping)
Line 152 and 183, data now shown
-> Two data from our preliminary test were not shown here but the as explained in the text, the tests showed diverse amplication patterns, so we used RAPD analysis for this study.
Results
Line 164, how can it be explained that none of the isolates from stool samples of humans suffering food poisoning contains the sed gene, whereas SEA and SED are the most common toxins in staphylococcal food poisoning outbreaks (line 37 and references 2 and 3)?
-> Various types of enterotoxin genes from S. aureus are related to food poisoning and the explanation for this result is explained in the discussion part. (L214-215:This apparent discrepancy may reflect differences in number and origin of the samples/isolates, and the composition of our multiplex PCR assay.)
Line 169, moderately resistant, why is only erythromycin mentioned here? I think all antibiotics other than ampicillin, penicillin and rifampin have to be mentioned here (moderate resistance, 31 – 78 %).
-> Changed the sentence as suggested (moderate resistance antimicrobial agents are listed; cefepime, cefotetan, chloramphenicol, ciprofloxacin, clindamycin, erythromycin, gentamicin, imipenem, oxacillin, tetracycline, and trimethoprim/sulfamethoxazole)
Line 175, please delete current text often
-> The table name is corrected and deleted the guideline text. (Added: Antimicrobial-agent resistance patterns of the 95 S. aureus strains. Deleted: This is a table. Tables should be placed in the main text near to the first time they are cited.)
Discussion
Line 215, how can the absence of a correlation be explained with the composition of the multiplex PCR method?
-> The absence of a correlation can be explained with number and origin of the samples/isolates. The composition of the multiplex PCR can be expanded for the detection of more diverse enterotoxin genes for the better understanding of the prevalence of enterotoxin genes in the field but deleted the expression not to confuse the readers.
Line 222, MALDI-TOF MS
-> The word is changed as suggested (MALDI-TOF -> MALDI-TOF MS).
Conclusions
Line 240, We analysed
-> The word is changed as suggested (analyzed -> analysed).
Line 245, illnesses.
-> The period is added.
References
Please indicated the year in bold, and please indicate the abbreviated journal name in italic
Supplementary material
Table S1, text below table: References; please indicate only the journal volume, not the issue.
-> The references are corrected as suggested.